# Sonographic Features Differentiating Follicular Thyroid Cancer from Follicular Adenoma–A Meta-Analysis

**DOI:** 10.3390/cancers13050938

**Published:** 2021-02-24

**Authors:** Martyna Borowczyk, Kosma Woliński, Barbara Więckowska, Elżbieta Jodłowska-Siewert, Ewelina Szczepanek-Parulska, Frederik A. Verburg, Marek Ruchała

**Affiliations:** 1Department of Endocrinology, Metabolism and Internal Medicine, Poznan University of Medical Sciences, 61-701 Poznan, Poland; kosma@ump.edu.pl (K.W.); jodlela@wp.pl (E.J.-S.); ewelina@ump.edu.pl (E.S.-P.); mruchala@ump.edu.pl (M.R.); 2Department of Computer Science and Statistics, Poznan University of Medical Sciences, 61-701 Poznan, Poland; barbara.wieckowska@ump.edu.pl; 3Department of Radiology and Nuclear Medicine, Erasmus MC, 3015 Rotterdam, The Netherlands; f.verburg@erasmusmc.nl

**Keywords:** thyroid, ultrasonography, follicular neoplasm, follicular lesion of unknown significance, follicular thyroid cancer

## Abstract

**Simple Summary:**

The risk of thyroid malignancy assessment may include certain ultrasound features. The analysis is lacking for the differentiation of follicular thyroid adenomas and cancers (FTAs and FTCs). Our meta-analysis aimed to identify sonographic features suggesting malignancy in the case of follicular lesions, potentially differentiating FTA and FTC. Based on twenty studies describing sonographic features of 10,215 nodules, we found that the most crucial feature associated with an increased risk of FTC were tumor protrusion (odds ratios—OR = 10.19), microcalcifications or mixed type of calcifications: 6.09, irregular margins: 5.11, marked hypoechogenicity: 4.59, and irregular shape: 3.6.

**Abstract:**

Certain ultrasound features are associated with an increased risk of thyroid malignancy. However, they were studied mainly in papillary thyroid cancers (PTCs); these results cannot be simply extrapolated for the differentiation of follicular thyroid adenomas and cancers (FTAs and FTCs). The aim of our study was to perform a meta-analysis to identify sonographic features suggesting malignancy in the case of follicular lesions, potentially differentiating FTA and FTC. We searched thirteen databases from January 2006 to December 2020 to find all relevant, full-text journal articles written in English. Analyses assessed the accuracy of malignancy detection in case of follicular lesions, potentially differentiating FTA and FTC included the odds ratio (OR), sensitivity, specificity, positive and negative predictive values. A random-effects model was used to summarize collected data. Twenty studies describing sonographic features of 10,215 nodules met the inclusion criteria. The highest overall ORs to increase the risk of malignancy were calculated for tumor protrusion (OR = 10.19; 95% confidence interval: 2.62–39.71), microcalcifications or mixed type of calcifications (coexisting micro and macrocalcifications): 6.09 (3.22–11.50), irregular margins: 5.11 (2.90–8.99), marked hypoechogenicity: 4.59 (3.23–6.54), and irregular shape: 3.6 (1.19–10.92). The most crucial feature associated with an increased risk of FTC is capsule protrusion, followed by the presence of calcifications, irrespectively of their type.

## 1. Introduction

Ultrasound-guided fine-needle aspiration biopsy (FNAB) is a widely used procedure and a gold standard for the evaluation of thyroid nodules [1]. However, despite its utility, it has certain limitations, particularly when it comes to follicular lesions [2]. Then the cytological diagnosis is often consistent with “atypia of undetermined significance” (AUS) or “follicular lesion of undetermined significance” (FLUS), the III diagnostic category of the Bethesda System for Reporting Thyroid Cytopathology, or IV diagnostic category being follicular neoplasm or suspicion of follicular neoplasm [3]. The malignancy risk for the III category is estimated at 10–30%, while it is slightly higher in the IV category, being equal to 25–40% [3]. However, the risk may differ according to the population studied, i.e., in previously iodine-deficient countries, the estimated malignancy risk for these categories may be 2.4–5.2% and 8.2–19%, respectively [4]. Therefore, it is of considerable significance to find accessible tools or criteria that would allow distinguishing between benign and malignant lesions in case of inconclusive biopsy results. The estimation of the malignancy risk preoperatively is of enormous importance as it allows doctors to decide on surgical treatment or follow-up.

Despite increasing accessibility of novel imaging methods, e.g., positron emission tomography with computed tomography, they were not demonstrated to result in a dramatic reduction of unnecessary thyroidectomies performed among patients with FNAB Bethesda IV category. Another option is the identification of particular genetic markers obtained from cytological material [2]. However, results of genetic studies so far have not yielded satisfactory sensitivity and specificity while still being an invasive procedure, considerably expensive, and not widely available [5]. Unlike them, thyroid ultrasound is nowadays a routine examination, which is quick, non-invasive, cheap, and reproducible [6]. Ultrasound features could potentially be used to stratify the risk of malignancy in Bethesda III and IV categories. According to the results of several research and meta-analyses, there are certain ultrasound features associated with increased risk of malignancy [7,8]. Among them, the most useful were “taller than wide shape”, decreased elasticity, irregular margins, microcalcifications, lack of halo, and hypoechogenicity [7,9,10]. However, these concern mainly the most common type of thyroid neoplasm-papillary thyroid cancer (PTC), i.e., two large meta-analyses by Brito et al. and Wolinski et al. took into account all types of thyroid cancer, but with definite predominance (84% and 89%, respectively) of PTC [7,9]. Still, little is known about the features of other thyroid cancer types, i.e., follicular (FTC) or medullary thyroid cancer (MTC). We hypothesize that conclusions drawn from meta-analyses taking into account in majority PTCs cannot be extrapolated and used for the estimation of malignancy risk of FTCs or MTCs. There was one meta-analysis published to date, aiming to summarize the characteristics of the ultrasound picture of MTCs [11]. However, to the best of the authors’ knowledge, there has been no meta-analysis concerning ultrasound features indicating FTC. It has already been observed that PTCs and FTCs differ in terms of size, contour” and echogenicity of the lesion evaluated preoperatively by conventional ultrasonography [12]. There were only a few studies devoted to sonographic characteristics of FTC [13,14,15]. Other studies report the sonographic features of thyroid lesions according to the exact histopathological diagnosis, instead of only distinguishing between benign and malignant lesions, and include, among other FTCs and follicular thyroid adenomas (FTAs). However, they represent a limited number of follicular lesions; indicated sonographic features vary greatly and may not be useful in the differentiation of follicular lesions [16,17,18,19]. Another promising method potentially differentiating FTA and FTC are elastography and tridimensional Doppler [20,21]. Our study aimed to perform a meta-analysis of so far conducted studies and identify sonographic features suggesting malignancy in the case of follicular lesions, potentially differentiating FTA and FTC.

## 2. Results

After a complete systematic review was performed, 20 studies met the inclusion criteria. They covered analyses of 10,215 nodules. The search results and steps of selection are shown in the flowchart (Figure 1 and Table 1). The overall odds ratios for particular features giving a risk of FTC varied from 1.44 to 10.19 (Table 2 and Figure 2).

Specificity to predict FTC for individual features varied from 18% to 100%, and the sensitivity ranged from 3% to 93%. Negative predictive value (NPV) was 64% to 90%, and positive predictive value (PPV) was 28% to 96% (Table 2 and Figure 2). All tables in the Appendix A present the pooled estimates of sensitivity, specificity, PPV, NPV, and odds ratios obtained from the bivariate model.

The highest overall odds ratio in increasing the risk of malignancy was calculated for tumor protrusion odds ratio (OR) (95% confidence interval (CI)) = 10.19 (2.62–39.71), microcalcifications or mix type of calcifications (micro and macrocalcifications): 6.09 (3.22–11.50), irregular margins: 5.11 (2.90–8.99), marked hypoechogenicity: 4.59 (3.23–6.54), and irregular shape: 3.60 (1.19–10.92). The lowest OR was characteristic for a Doppler pattern of three or more: 1.44 (0.76–2.74).

The highest overall sensitivity with its 95% CI was 93% (87–97%) for solid or mainly solid structure, and the lowest was for taller than wide size: 3% (0–10%). The highest specificity was 100% (99–100%) for tumor protrusion, and the lowest was for solid or mainly solid structure: 18% (6–31%). Accordingly, the highest PPV was 96% (70–100%) for tumor protrusion; the lowest was for solid or mainly solid structure 28% (20–37%) and for a Doppler pattern of three or more 28% (10–51%). Furthermore, the highest NPV was 90% (81–96%) for solid or mainly solid structure, and the lowest was for tumor protrusion: 64% (61–68%). Table 2 and Appendix A show detailed calculations of OR, sensitivity, specificity, PPV, and NPV for all analyzed features with their overall summaries, tests of heterogeneity, and Egger’s asymmetry tests.

Patients finally diagnosed with FTC were more than 10 times more likely to have a tumor protrusion (Figure 3)—OR (95% CI): 10.19 (2.62–39.71) (Appendix A). The analysis included 633 patients and the group proved to be homogenous (test for heterogeneity: *I*^2^ = 0.0%, *p* = 0.4350. The specificity (95% CI) of this feature reached 1.00 (0.99–1.00) with low sensitivity (95% CI) of 0.06 (0.03–0.09).

The analysis of microcalcifications or mixed type (coexisting micro and macrocalcifications) was based on nine publications covering 1199 patients (Appendix A). No recent studies were located, and the group proved to be homogeneous (test for heterogeneity: *I*^2^ = 0%, *p*-value = 0.5494) and publication bias was not reported (Egger’s *p*-value = 0.0800). Summary OR, presented as an overall OR (95% CI) = 6.09 (3.22–11.50), meaning that cancer patients are more than six times more likely to have a positive microcalcifications or mixed type (coexisting micro and macrocalcifications) than those with adenoma. There was relatively low overall sensitivity (95% CI) = 0.10 (0.03–0.19) and overall PPV (95% CI) = 0.53 (0.19–0.86) but quite high overall specificity (95% CI) = 0.97 (0.95–0.99) and overall NPV (95% CI) = 0.78 (0.69–0.88). Overall sensitivity and specificity determined jointly in the hierarchical summary receiver operating characteristic (HSROC) model was similar and amounted to overall sensitivity (95% CI) = 0.1 (0.04–0.21), overall specificity (95% CI) = 0.98 (0.95–0.99).

For irregular margins, initially, the analysis covered 14 papers and a total of 1721 patients (Appendix A). Studies were distributed symmetrically (Egger’s *p*-value = 0.0980). Strong study heterogeneity was detected (*I*^2^ = 75.3%. *p* < 0.0001) which resolved after excluding four outlier studies. As a result, outliers were identified based on sensitivity analysis and funnel plot inspection. After removal of the outlier, the group was more homogeneous (*I*^2^ = 36.59%, *p* = 0.1156) and based on 1227 patients. Originally overall OR (95% CI) was 3.49 (1.66–7.35) and after exclusion of the indicated study, the summary OR increased to the value of overall OR (95% CI) = 5.11 (2.90–8.99). Overall sensitivity (95% CI) was quite low = 0.24 (0.13–0.37) and overall NPV (95% CI) = 0.80 (0.74–0.86) and overall specificity (95% CI) was high = 0.94 (0.90–0.96) and overall PPV (95% CI) was = 0.53 (0.34–0.71). High specificity and quite low sensitivity were also confirmed by the HSROC curve analysis: overall specificity (95% CI) = 0.94 (0.89–0.96), HSROC overall sensitivity (95% CI) = 0.24 (0.15–0.37).

For hypo- and markedly hypoechogenic structure, initially, the analysis covered 16 papers and a total of 1864 patients (Appendix A). Studies were distributed symmetrically (Egger’s *p*-value = 0.2811). Strong study heterogeneity was detected (*I*^2^ = 68.1%, *p* < 0.0001) which resolved after excluding two outlier studies. As a result, outliers were identified based on sensitivity analysis and funnel plot inspection. After removal of the outlier, the group was more homogeneous (*I*^2^ = 34.96%, *p* = 0.0955) and based on 1610 patients. Originally overall OR (95% CI) was 3.69 (2.30–5.92) and after exclusion of the indicated study, the summary OR increased slightly to the value of overall OR (95% CI) = 4.59 (3.23–6.54). High overall sensitivity (95% CI) = 0.74 (0.60–0.86) and overall NPV (95% CI) = 0.87 (0.81–0.92) and quite high overall specificity (95% CI) = 0.63 (0.53–0.73) and overall PPV (95% CI) = 0.44 (0.35–0.53). High sensitivity and specificity values were also confirmed by the HSROC curve analysis: overall sensitivity (95% CI) = 0.74 (0.62–0.84), HSROC overall specificity (95% CI) = 0.63 (0.53–0.73).

## 3. Discussion

The incidence of differentiated thyroid cancer (DTC) has risen considerably over the past few decades. It is attributed mostly to the increasing rate of PTC, which constitutes the primary histological type of thyroid cancer [4]. The exact data on the changing rate of FTC is unavailable. However, American studies demonstrated an increase of 30% in the follow-up period from 1980 to 2009 [38]. On the other hand, the incidence of FTC was found to be reduced with the introduction of the iodination program in the previously iodine-deficient areas [39]. However, it may still account for up to 20% of differentiated thyroid cancers in the regions previously affected by iodine deficiency, constituting an important clinical problem.

The issue of sonographic features of malignancy has been covered in a few large meta-analyses. Brito et al., in their meta-analysis covering 31 studies including 18,288 focal lesions, indicated that the best predictor of malignancy was the shape of the lesion; “taller than wide” lesions were 11 times more likely to be diagnosed with thyroid cancer than those oval or round. The second important ultrasound feature that was most strongly associated with malignancy risk was the presence of microcalcifications (OR = 6.8) [9]. The size of the lesion did not correlate importantly with malignancy risk. On the other hand, the authors indicated that spongiform appearance and the presence of a cystic component were significantly associated with the benignity of a lesion. In another meta-analysis by Campanella et al., again, the shape of the lesion was found to correlate with malignancy risk (OR = 10.2). Other but less suspected features were lack of halo, presence of microcalcifications, and irregular borders [10]. According to recent European Thyroid Association guidelines, lesions presenting at least one of the following features: shape different than oval, irregular borders, microcalcifications, and deep hypoechogenicity, were at the highest malignancy risk, equal to 26–87%. The more malignancy features are present in the lesion, the highest malignancy risk is. This approach allows for the identification of thyroid cancer with high specificity at the level of 83–84% and moderate sensitivity equal to 26–59% [40]. Moreover, incomplete calcified capsule, thick halo, dominant central vascularization, and decreased elasticity of the lesion, increase the risk of moderately suspected lesions. On the other hand, thin halo, cystic component, comet-tail artifacts, peripheral vascularization, and high elasticity of the lesion were found to decrease malignancy risk. The results of a meta-analysis, including only prospective studies with histopathological verification previously performed by our team, were consistent with the findings as the most critical ultrasound feature associated with the highest malignancy risk (OR = 13.7) was the lesion shape [7]. Further essential features most strongly suggesting malignant character were decreased elasticity, irregular margins, and presence of microcalcifications. However, one must remember that in all of the mentioned meta-analyses, the predominant type of malignant lesions were PTCs. Moreover, many studies do not provide information on the histopathological type of thyroid cancer. In the studies in which the final histopathology is given, 89% of cancers were PTCs [7]. Thus, it is not clear whether the conclusions from these studies can be extrapolated on other types of thyroid cancers, i.e., follicular of medullary type. To the best of our knowledge, our research constitutes the first meta-analysis aiming to compare sonographic features differentiating FTC from follicular thyroid adenoma.

Our meta-analysis demonstrated that the sonographic feature the most strongly increasing the risk of FTC, but not underlined in the previous studies, was capsule protrusion. Although only two studies took into account this feature, it turned out to be the substantial differentiating factor between FTA and FTC, with an OR at the level of 10.19 [13,32]. Capsule protrusion towards the surrounding structures with or without visible capsule disruption can be considered as a risk factor for the extrathyroidal extension, which is equal to 61% in these subjects, while 31% for macroscopic invasion [40].

Many studies have identified the presence of calcifications as malignancy predictors. While microcalcifications are one of the features significantly associated with the diagnosis of PTC, our results demonstrated that malignancy of follicular lesion might be suggested by the presence of not only microcalcifications but also mixed calcifications of a different type. In our meta-analysis, the presence of entirely macrocalcifications (>1 mm) was associated with a moderate risk of FTC with ORs between 2–3. Quite similar results were obtained by Kunt et al., where authors aimed to identify the risk factors of malignancy in a group of nodules preoperatively diagnosed as suspicion of FTC, and intranodular calcifications increased by about three times the relative risk of malignancy when present [41]. The diagnostic utility of calcifications in the case of FTC is limited by its low sensitivity. In the study by Sillery et al. comparing the distribution of particular sonographic variables in 52 FTAs vs. 50 FTCs, the feature occurred only in 14% of FTCs [13]. However, the absence of calcifications may have a negative predictive value. In the study by Zhang et al., over 90% of FTAs did not present calcifications, while the diagnosis of FTC was more frequently associated with the presence of calcifications (not only microcalcifications but also macrocalcifications and peripheral type). Still, this was not a sensitive feature, as in 55.5% of FTCs, calcifications were absent [14]. In a Chinese group, punctuate calcifications were more prevalent in FTCs (40.5%) compared to 13.5% of FTAs [23]. In the study by Kuo et al., either type of calcification was present in about one-third of FTCs, compared to only 3.6% of FTAs [36], while Liu et al. noted that macrocalcifications were the type of calcifications most importantly differentiating FTCs from FTAs, with specificity equal to 90.3% [37].

Another essential feature confirmed to be associated with FTC risk was a solid character of a lesion as well as heterogeneous and hypoechogenic echostructure. Hypoechogenicity was the most frequent ultrasound feature, occurring in 82% of FTCs reported by Sillery et al. [13]. In another study, by Chng et al., evaluating lesions diagnosed as follicular neoplasm on cytology, hypoechogenicity was present in 74.3% of FTCs vs. 51.4% of FTAs [42], and 64.9% vs. 39.2%, respectively in a group by Lai et al. [23]. The latter group also reported that the absence of cystic component was more frequently associated with FTC than FTA (78.4% vs. 54.1%) [23]. Predominant (>50%) cystic component was a predictor of benignity and presence of FTA in the group by Sillery et al. [13]. Authors explain that hypoechogenicity and lack of cystic degeneration might be a consequence of the rapidity of growth of the tumor cells, resulting in a disturbed formation of follicles, more typical for malignant lesions [13]. In another study by Zhang et al., a previous observation was confirmed that cystic component was significantly more frequent prevalent in FTAs, while in all of the studied 36 FTCs, a cystic component comprised less than 25% of the nodule volume [14]. Another FTC feature confirmed in this study was hypoechogenicity, while other echogenicity shades were more typical of FTAs. Most FTCs (83.3%) presented with heterogeneous echogenicity, while 80.8% of FTAs characterized by homogeneous echotexture. Authors demonstrated that a predominantly solid pattern, a heterogeneous echogenicity, and presence of calcifications were factors independently associated with the risk of FTC. The observations were consistent with the results obtained by Seo et al. [15]. Their logistic regression analysis demonstrated that predominantly solid character, mixed echotexture, and presence of microcalcifications or rim calcifications significantly increased the relative risk for FTC. However, neither Kuo et al. nor Liu et al. found significant difference in terms of nodule composition between FTAs and FTCs [36,37]. In addition, Liu et al. demonstrated that FTCs are more often hypoechogenic, while FTAs isoechogenic or presenting mixed echogenicity [37].

Irregular (microlobulated or spiculated) margins [40] increased the malignancy rate by 2.92, according to our results. The study by Maia et al. aiming to evaluate the value of ultrasound retrospectively to predict malignancy in indeterminate thyroid nodules by cytology confirmed this observation. Multivariate analysis revealed that borders irregularity on sonographic examination predicted malignancy risk in indeterminate thyroid nodules with 76.9% accuracy [43]. In another study by Chng et al., evaluating lesions diagnosed as follicular neoplasm on cytology, irregular margins were found to be present in 20% of FTCs but no FTA [42]. The irregular margin was also one of the features more prevalent in FTCs (21.6%) vs. 1.4% FTAs in a group by Lai et al. [23]. Both Liu et al. and Kuo et al. found that spiculated, lobulated, or irregular margins were significantly more prevalent in FTCs, while FTAs presented with a rather smooth contour [36,37].

The characteristic “taller than wide” shape of a lesion, so strongly associated with malignancy rate if PTCs are concerned, does not seem to play an important role in the case of FTCs. Our results demonstrated that OR for this feature was equal to 2.81. In another study by Chng et al., evaluating lesions diagnosed as follicular neoplasm on cytology and taller than wide morphology was not very frequent in FTCs (17.1%) but occurred rarer in FTAs (0.9%) [42]. In the studies by Liu et al. and Kuo et al., the taller than wide shape was not a very important feature useful in differentiation between FTAs and FTCs [36,37].

An OR between 2–3 was yielded for lack of halo or presence of thick halo and solitary lesions. Our conclusions about the halo sign are consistent with the risk factors for thyroid malignancy in general. Recent European Thyroid Association Guidelines indicate that a thin halo decreases the malignancy risk by about three-times (OR = 0.3), while thick or lack of halo increase the malignancy risk, with ORs equal to 3.4 and 7.1, respectively [10,40,44]. Halo was not present in 64% of FTCs in a group reported by Sillery et al. [13], being the second (after hypoechogenicity) most common feature associated with the malignant follicular lesion. The presence of halo sign may be attributed to the preserved capsule of FTA, which continuity is a feature allowing pathologists to differentiate between FTA and FTC. The presence of a thin halo was almost three times more frequently observed in FTAs in comparison to FTCs in a study by Zhang et al., while incomplete or unevenly thick halo was a feature significantly more frequently occurring in FTCs [14]. In the Chinese study, the authors also noticed the almost twice more common absence of halo in FTC patients (67.6%) vs. the FTA group (36.5%) [23].

Less important feature suggesting the malignant character of follicular lesions in our meta-analysis was size > 4 cm. The median volume of FTC (11.75 mL) was larger than FTA (5.95 mL) in the study by Sillery et al. [13]. Previous studies comparing ultrasound features of lesions eventually diagnosed as PTC or FTC demonstrated that FTCs were significantly greater than PTCs [12,45]. This may be explained by the hypothesis also supported by some genetic studies [46], that FTC may result from the transformation from FTA and by the difficulties in cytological detection of malignant features in small FTC tumors [12]. Other studies did not report a significant difference in terms of size between FTAs and FTCs [36,37].

One of the less useful features of FTCs in our meta-analysis was the presence of central vascularization. The vascularization pattern on the Color Doppler examination was not a helpful feature in the differentiation of FTC and FTA by Sillery et al. [13]. In another study comparing ultrasound features of 37 FTCs with 74 FTAs, the incidence of intranodular vascularization did not differ significantly between the two groups [23]. This feature was also of limited value in the prediction of malignancy in the case of PTCs [7]. However, the reported results are not entirely consistent, as Kunt et al. indicated that intranodular vascularization (Doppler pattern three for a peripheral ring of flow and a small-to-moderate amount of internal flow, and four for extensive internal flow with or without a peripheral ring) [47], as the most useful predictor of malignancy with an OR at the level of 14.7, which is in contrast to our and previous observations [41].

Once sonoelastography was introduced to thyroid diagnostics, it raised hopes that it would be of value in presurgical and non-invasive differentiation of follicular lesions. Fukunari et al. analyzed 56 follicular lesions. Out of 51 FTAs, 48 (94.1%) presented with normal elasticity, while all FTCs demonstrated a characteristic pattern of elasticity, corresponding with an elastic central part and a stiff peripheral region. The authors concluded that sonoelastography might reflect the differences in the histopathological structure of follicular lesions and might be helpful in differentiation between benign and malignant follicular lesions. Another communication from this research group resulted in the conclusion that over 70% of FTCs present with such a sonoelastographic pattern [48]. Another paper by Rago et al. also postulated the potential usefulness of sonoelastography in the presurgical prediction of the character of thyroid lesions, in which cytological examination yielded inconclusive results [49]. However, future studies did not confirm the previous findings and usefulness of sonoelastography of differentiation of follicular lesions. As there was only one full-text paper encompassing the sonoelastographic picture of follicular lesions, we were not able to include this feature in our meta-analysis. In the study by Liu et al., the speed of shear waves propagation on sonoelastographic examination was greater for FTCs if compared to FTAs [37].

The most crucial feature associated with an increased risk of FTC is capsule protrusion, followed by the presence of calcifications, irrespectively of their type. The most important ultrasound malignancy risk factors for PTC were rather taller than wide (ORs = 13.7, 11.4, and 10.15), which was only the tenth feature in our analysis for FTC with an OR = 2.52. However, microcalcifications and irregular margins seem to be common malignancy ultrasound features both for FTC and PTC [7,9,50].

Currently, due to the inconsistency of ultrasound terminology and to enable easier risk of malignancy determination for thyroid nodules ultrasound assessment, there are many risk stratification models comprising conventional ultrasound and elastography characteristics. They enable a better combined evaluation of thyroid nodules and are considered an important step in endocrinology [40,51,52,53,54,55]. Although helpful in the assessment of cytologically equivocal thyroid nodules, according to some studies [56], they may have limited clinical values for risk stratification of intermediate cytological results according to the others [57]. Therefore, there is still a need for research in this field.

## 4. Materials and Methods

### 4.1. Search Strategy

We carried out the meta-analysis following the guidelines formulated in the Cochrane Handbook for Systematic Reviews of Interventions and the Preferred Reporting Items for Systematic Reviews and Meta-Analyses (PRISMA) guidelines [58]. We searched PubMed, MEDLINE, Academic Search Complete, CINAHL Complete, CINAHL, Scopus, Cochrane, Health Source: Nursing/Academic Edition, Web of Knowledge, MasterFILE Premier, Health Source-Consumer Edition, Agricola, Dentistry and Oral Science Source databases from January 2006 up to December 2020 to find all relevant, full-text journal articles written in English.

We included studies, regardless of their sample size, with the investigation of the association between one or more ultrasound feature and the risk of follicular thyroid malignancy, which did not have any restriction criteria for the inclusion of detected nodules in the study, such as nodule size or thyroid-stimulating hormone (TSH) levels [10]. We considered histopathological diagnosis after surgery to be the gold standard reference test and included only studies considering the histopathological result of FTA and FTC as the exclusive diagnoses, as well as within an analysis of different thyroid histopathological diagnoses. Studies were excluded if focusing only on particular subgroups of patients such as pediatric patients only, with a prior history of thyroid cancer or were clearly exposed to known risk factors for thyroid cancer, e.g., Chernobyl survivors or particular types of nodules (e.g., palpable, less than 1 cm, pure cystic or solid, etc.) [7,9,50].

The search strategy included Medical Subject Headings terms and keywords: “thyroid and (“follicular cancer” or “follicular carcinoma” or “follicular neoplasm” or “follicular adenoma” or “follicular nodule”) and (ultrasound or ultrasonography or elastography or “color doppler” or “power doppler”)”. Reference lists of all the selected articles, previous meta-analyses, and reviews were hand-searched for any additional articles.

### 4.2. Data Extraction

Two authors (M.B. and E.J.S.) independently selected papers, which fulfilled the inclusion criteria and extracted data for the outcomes using a standardized data extraction form. Relevant data included articles assessing echogenicity, calcifications, presence of a “halo”, size, shape, protrusion, margins, Doppler pattern, solitarity, and structure of nodules. Another author (E.S.P.) rechecked the extracted data.

### 4.3. Assessment of Methodological Quality

The risk of bias in the included studies was independently assessed by two authors (MB and ESP by the Cochrane risk of bias tool [20]. As recommended for diagnostic accuracy-test studies, the revised Quality Assessment of Diagnostic Accuracy Studies-2 (QUADAS-2) tool was also used. All included studies were assessed using the Newcastle-Ottawa Scale [59]. Studies with a result of seven stars or more were included.

### 4.4. Statistical Analysis

Analyses assessing the accuracy of malignancy detection in case of follicular lesions, potentially differentiating FTA and FTC included the odds ratio (OR), sensitivity, specificity, positive predictive values (PPV), and negative predictive values (NPV). A random-effects model described by DerSimonian and Laird was used to summarize collected data.

In the first stage, we calculated ORs and assessed studies’ heterogeneity and publication bias. Statistical heterogeneity between the studies was examined using Cochrane’s Q statistics and *I*^2^ statistics. The publication bias was explored by visual inspection of funnel plots, and asymmetry was tested formally with Egger’s regression test [60,61]. Furthermore, a sensitivity analysis was performed for parameters showing significant heterogeneity. In the case of high heterogeneity (i.e., *I*^2^ > 50% and *p* < 0.05), outlying studies were identified. The meta-analysis was repeated to confirm the obtained results, excluding outliers and the overall OR, and the heterogeneity test results were given again. In the event of a zero outcome, continuity correction was performed by adding a correction factor of 0.5.

In the second stage, after outliers exclusion, based on the number of true positive (TP), true negative (TN), false positive (FP), and false negative (FN) results univariates of sensitivity, specificity, Negative Predictive Value (NPV), and Positive Predictive Value (PPV) with 95% confidence intervals were estimated using the exact binomial Clopper-Pearson method. In the meta-analysis, the pooled estimation was calculated after Freeman-Tukey Double Arcsine Transformation to stabilize the variances [62]. Additionally, for the analysis of traits that were based on more research and met the assumptions of the HSROC model (currently recommended by the Cochrane Collaboration), bivariate meta-analyses were performed to jointly models both sensitivity and specificity.

The significance level *p* = 0.05 was assumed in all analyzes. The analysis of the odds ratio was carried out in the PQStat v1.6.6 program, while the results regarding sensitivity, specificity, PPV, and NPV were obtained in the Stata v14 package, using the metaprop and metandi functions.

## 5. Conclusions

In conclusion, sonographic features associated with the malignancy of follicular lesions are distinct from those widely reported for all thyroid cancers, of which the predominant histological type is PTC. The most crucial feature associated with an increased risk of FTC is capsule protrusion, followed by the presence of calcifications, irrespective of their type. Less specific but more frequent are the irregular shape of the lesion, solid character of the lesion, and hypoechogenicity. On the other hand, a high probability of a diagnosis of FTA is suggested by an oval or round shape of the lesion and the presence of a cystic component. Less specific features suggesting benign lesions are a lack of calcifications and a visible halo.

## Figures and Tables

**Figure 1 cancers-13-00938-f001:**
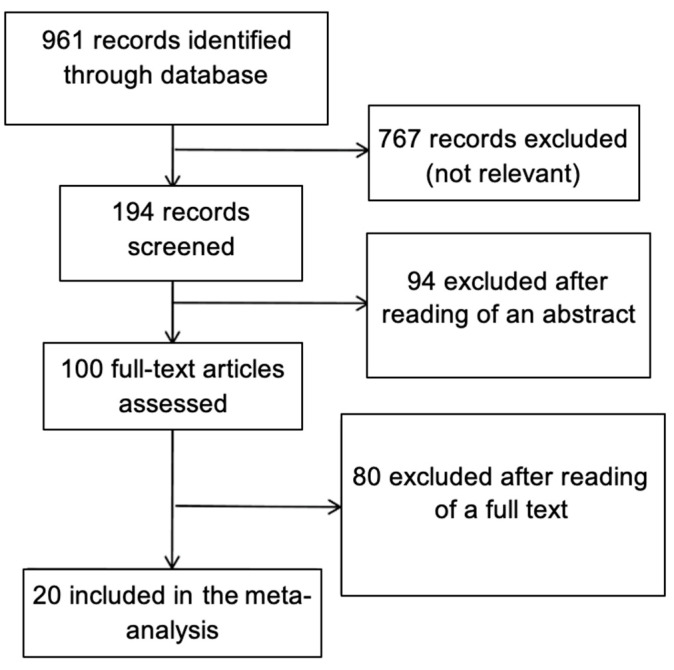
Methodological flow diagram of the meta-analysis for sonographic features differentiating follicular thyroid cancer from follicular adenoma utility.

**Figure 2 cancers-13-00938-f002:**
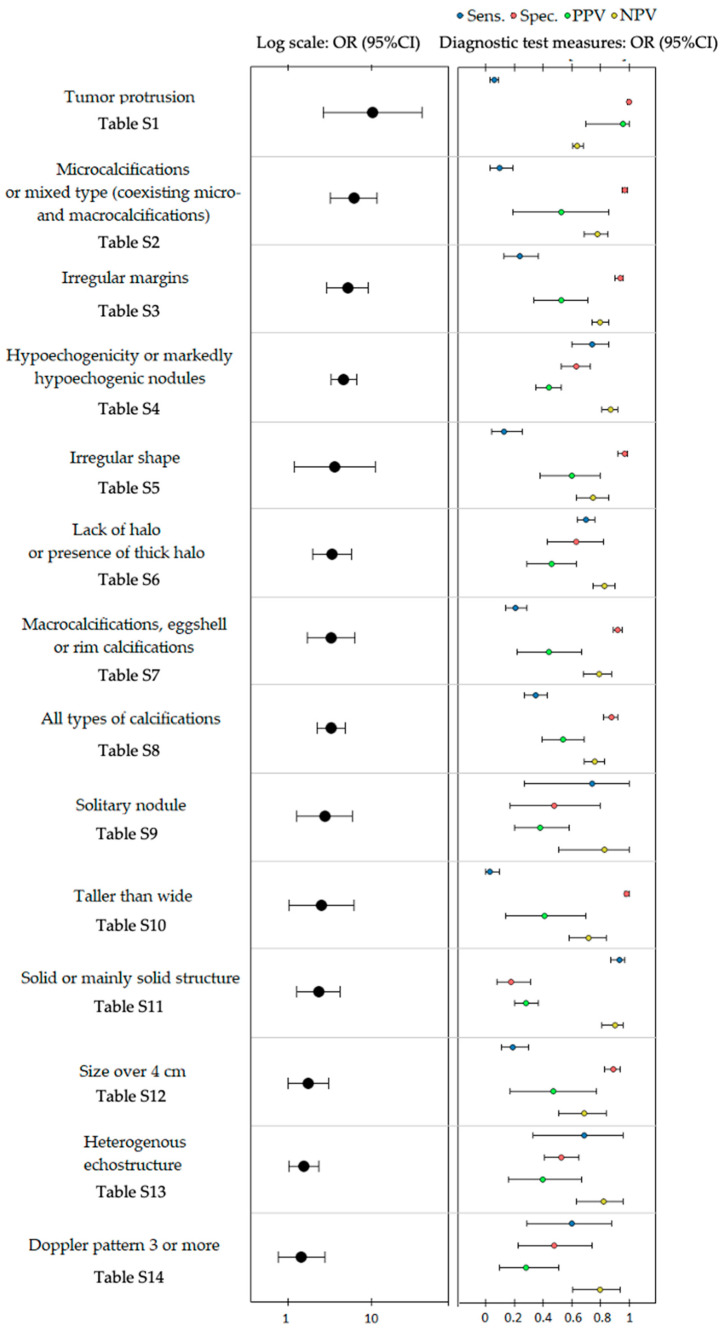
The graphic presentation of overall specificity, sensitivity, negative prognostic value (NPV), and positive predictive value (PPV) for sonographic features differentiating follicular thyroid cancer from follicular adenoma utility and their overall odds ratios (OR) with their 95% confidence intervals (95% CI).

**Figure 3 cancers-13-00938-f003:**
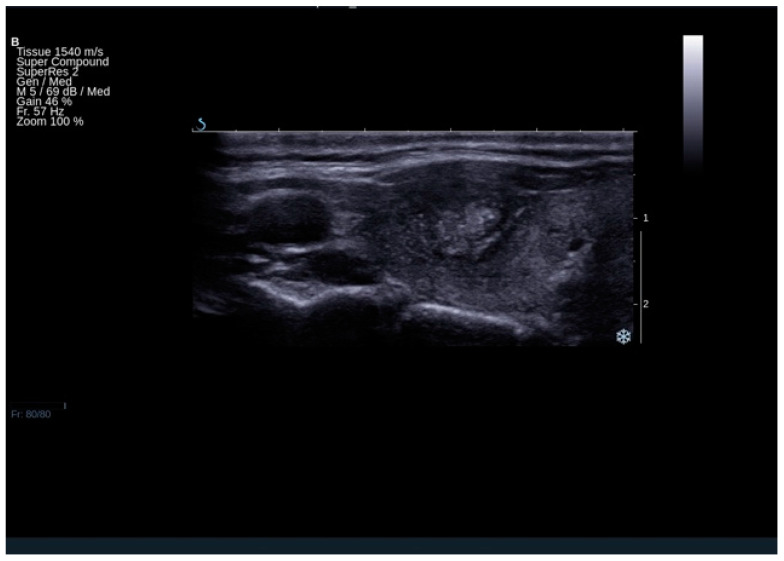
The result of ultrasound examination demonstrating thyroid lesion, which turned out to be follicular cancer on histopathological examination. The lesion presents tumor protrusion, irregular margins, microcalcifications, and heterogeneous echostructure.

**Table 1 cancers-13-00938-t001:** The list of included studies.

Author	Year	Number of Nodules (FTC/FTA); Malignancy Rate (%)	1	2	3	4	5	6	7	8	9	10	11	12	13	14
Seo HS et al. [15]	2009	126 (66/60) 52.4														
Sillery JC et al. [13]	2010	102 (50/52); 49.0														
Lee EK et al. [22]	2012	110 (33/77); 30.0														
Lai X et al. [23]	2013	111 (37/74); 33.3														
Lee KH et al. [24]	2013	75 (11/64); 14.7														
Lee SH et al. [25]	2013	66 (16/50), 24.2														
Pompili G et al. [26]	2013	102 (14/88); 13.7														
Kamran SC et al. [27]	2013	7348 (927/6421); 12.6														
Tutuncu J et al. [28]	2014	88 (6/82); 6.8														
Cordes M et al. [12]	2014	57 (24/33); 42.1														
Yoon JH et al. [29]	2014	177 (25/152); 14.1														
Zhang JZ et al. [14]	2014	88 (36/52); 40.9														
Cordes M et al. [30]	2016	200 (100/100); 50														
Jeong SH et al. [31]	2016	178 (22/156); 12.4														
Kobayashi K et al. [32]	2016	531 (184/347); 34.7														
Yang GCH et al. [33]	2016	279 (6/273); 2.2														
Kuru B.et al. [34]	2018	139 (51/88). 36.7														
Kim M et al. [35]	2018	160 (50/110); 31.3														
Kuo TC et al. [36]	2020	188 (49/139); 26.1														
Liu BJ et al. [37]	2020	90 (28/62); 31.1														

1-Tumor protrusion; 2-Microcalcifications or mixed type (coexisting micro- and macrocalcifications); 3-Irregular margins; 4-Hypoechogenicity or marked hypoechogenicity; 5-Irregular shape; 6-Lack of halo or presence of thick halo; 7-Macrocalcifications, eggshell or rim calcifications; 8-All types of calcifications; 9-Solitary nodule; 10-Taller than wide; 11-Solid or mainly solid structure; 12-Size over 4 cm; 13-Heterogenous echostructure; 14-Doppler pattern three or more. FTC: follicular thyroid cancer; FTA: follicular thyroid adenoma.

**Table 2 cancers-13-00938-t002:** Overall specificity, sensitivity, negative prognostic value (NPV), and positive predictive value (PPV) for sonographic features differentiating follicular thyroid cancer from follicular adenoma utility and their overall odds ratios (OR) with their 95% confidence intervals (95% CI).

Sonographic Feature	OR (95% CI)	Sensitivity (95% CI)	Specificity (95% CI)	PPV (95% CI)	NPV (95% CI)
Tumor protrusion Appendix A	10.19 (2.62–39.71)	0.06 (0.03–0.09)	1.00 (0.99–1.00)	0.96 (0.7–1.00)	0.64 (0.61–0.68)
Microcalcifications or mixed type (coexisting micro- and macrocalcifications) Appendix A	6.09 (3.22–11.5)	0.10 (0.03–0.19)	0.97 (0.95–0.99)	0.53 (0.19–0.86)	0.78 (0.69–0.85)
Irregular margins Appendix A	5.11 (2.9–8.99)	0.24 (0.13–0.37)	0.94 (0.90–0.96)	0.53 (0.34–0.71)	0.80 (0.74–0.86)
Hypoechogenicity or marked hypoechogenicity Appendix A	4.59 (3.23–6.54)	0.74 (0.6–0.86)	0.63 (0.53–0.73)	0.44 (0.35–0.53)	0.87 (0.81–0.92)
Irregular shape Appendix A	3.6 (1.19–10.92)	0.13 (0.04–0.26)	0.97 (0.92–0.99)	0.60 (0.38–0.8)	0.75 (0.63–0.86)
Lack of halo or presence of thick halo Appendix A	3.34 (1.95–5.73)	0.70 (0.64–0.76)	0.63 (0.43–0.82)	0.46 (0.29–0.63)	0.83 (0.75–0.90)
Macrocalcifications, eggshell or rim calcifications Appendix A	3.28 (1.69–6.35)	0.21 (0.14–0.29)	0.92 (0.89–0.95)	0.44 (0.22–0.67)	0.79 (0.68–0.88)
All types of calcifications Appendix A	3.26 (2.20–4.83)	0.35 (0.27–0.43)	0.88 (0.82–0.92)	0.54 (0.39–0.69)	0.76 (0.69–0.83)
Solitary nodule Appendix A	2.72 (1.26–5.86)	0.74 (0.27–1.00)	0.48 (0.17–0.80)	0.38 (0.20–0.58)	0.83 (0.51–1.00)
Taller than wide Appendix A	2.52 (1.02–6.19)	0.03 (0.00–0.10)	0.98 (0.97–1.00)	0.41 (0.14–0.70)	0.72 (0.58–0.84)
Solid or mainly solid structure Appendix A	2.3 (1.27–4.17)	0.93 (0.87–0.97)	0.18 (0.08–0.31)	0.28 (0.20–0.37)	0.9 (0.81–0.96)
Size over 4 cm Appendix A	1.73 (0.99–3.00)	0.19 (0.11–0.30)	0.89 (0.83–0.94)	0.47 (0.17–0.77)	0.69 (0.51–0.84)
Heterogenous echostructure Appendix A	1.53 (1.02–2.30)	0.69 (0.33–0.96)	0.53 (0.41–0.65)	0.4 (0.16–0.67)	0.82 (0.63–0.96)
Doppler pattern 3 or more Appendix A	1.44 (0.76–2.74)	0.60 (0.29–0.88)	0.48 (0.23–0.74)	0.28 (0.10–0.51)	0.80 (0.61–0.94)

## Data Availability

The data presented in this study are available in this article (and Appendix A).

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
