# Peer review of "Sonographic Features Differentiating Follicular Thyroid Cancer from Follicular Adenoma–A Meta-Analysis"

_cancers, 2021, doi:10.3390/cancers13050938_

Round 1

Reviewer 1 Report

The current manuscript is a meta-analysis reporting the data obtained by ultrasound evaluation in follicular lesions. The topic is of major importance, when looking into the prevalence of thyroid nodules in the general adult population with the constant increase in  the thyroid cancer cases.

REMARK 1

Introduction – considering the ultrasound characteristics regarding malignancy risk, there are also promising results of elastography and tridimensional Doppler improved presurgical risk stratification in cases with INTERMEDIATE CITOLOGY in FNAC:

Please do consider

  1. Andreea Borlea, Dana Stoian, Laura Cotoi, Ioan Sporea, Lazar Fulger, Ioana Mozos. Thyroid multimodal ultrasound evaluation – impact on presurgical diagnosis of intermediate cytology. Appl Sci. 2020, 10. doi: 3390/app10103439
  2. Dana Stoian, Florin Borcan, Izabella Petre, Ioana Mozos, Flore Varcus, Viviana Ivan, Andreea Cioca, Adrian Apostol, Cristina Adriana Dehelean. Strain elastography as a valuable diagnostic tool in intermediate cytology (Bethesda III) thyroid nodules. Diagnostics. 2019; 9:119
  3. Garino, F.; Deandrea, M.; Motta, M.; Mormile, A.; Ragazzoni, F.; Palestini, N.; Freddi, M.; Gasparri, G.; Sgotto, E.; Pacchioni, D.; et al. Diagnostic performance of elastography in cytologically intermediate thyroid nodules. Endocrine 2015, 49, 175–183. 

  4. Habib, L.A.M.; Abdrabou, A.M.; Geneidi, E.A.S.; Sultan, Y.M. Role of ultrasound elastography in assessment of indeterminate thyroid nodules. J. Radiol. Nucl. Med. 2016, 47, 141–147 

REMARK 2

Figure 4

  • Please do describe that you are referring to Shear Wave elastography. A clear description of the SWE values, ideally E Mean expressed in kPA, or the ration between the elasticity of the nodule and that of the normal thyroid parenchyma would be preferable to be used instead of the current image. Furthermore please do state that the normal thyroid parenchyma is not affected by autoimmune thyroid disease. Please do also describe the used probe and ultrasound machine used for presented the evaluation.

REMARK 3

Discussions

Currently there are many risk stratification models, used in evaluation of thyroid nodules, comprising 5 to 10 conventional ultrasound characteristics with or without information about elasticity of the nodules, the so called TIRADS models – you did not mentioned them at all. There are even data regarding which TIRADS model is better in risk stratification of thyroid nodules:

  1. Slowinska-Klencka D, Wysocka Konieczna K, Klencki M, Popowicz B. Diagnostic value of six TIRADS in cytologically equivocal thyroid nodules. J Clin Med 2020. (7):2281
  2. Chaigneau E, Russ G, Bigorgne C, et al…. TIRADS score is of limited clinical values for risk stratification of intermediate cytological results. Eur j Endocrinol 2018. 179, 13-20.

Reviewer 2 Report

The article is interesting and very well written. It aimed to identify sonographic features suggesting malignancy in case of follicular lesions.

Given the diffusion and the importance of ultrasonography in the study of thyroid nodules, this study gains importance and relevance.

Even if it is not the first study about this and it lacks of novelty, I think the meta-analysis is quite complete and well developed.

Reviewer 3 Report

Main points
1. The meta-analysis covers the literature up to July 2018 and is thus outdated. It should be updated to include all literature up the end of 2020.
2. In the end, most of the identified high-risk features are not really distinct from those of PTC. The claim made in the manuscript (line 22 and elsewhere) is therefore not supported by the data.
3. The inclusion and exclusion criteria should be listed. These are separate from the "search terms" and the procedure in Figure 1. Also, in Figure 1, "failed to present specific data" and "absence of key information" are not specific enough. Reference should be made to exclusion criteria.
4. In Figure 2, information seems to be missing at the end of the 3rd and 12th features.
4.1.1. Figures 2 and 3 should be merged and presented as a single Table. The graphics are useless, as is the background color (in fact, there is no need for color at all).
5. Lines 109-149: These 4 paragraphs discuss 4 features of nodules. Why were these 4 features specifically selected for detailed presentation among all the others?
6. Figure 4 is irrelevant, as elastography is not analyzed in the Results, only mentioned in the Discussion. It is also insufficiently explained, and the description in the legend does not seem to match the appearance of the image.
7. Line 132: "or mixed results". This refers to a different feature, with a lower OR.
8. Lines 195-196: The meaning is unclear.
9. About "capsule protrusion": This is not a standard feature according to EU-TIRADS or ATA. What does it mean? The tumor protruding out of its own capsule? Or out of the capsule of the thyroid gland? Images of capsule protrusion would be really useful to illustrate this feature (as opposed to the elastography image).
10. Lines 234-242: The terminology used to describe margins is confusing and misleading; "blurred", "irregular" and "indistinct" do not mean the same thing. The authors should refer to EU-TIRADS for a definition of suspicious margins (lobular or speculated).
11. Line 263: I don't think that we know for sure that FTC results from transformation of FAs. Can the authors provide a specific reference to support their strong statement?
12. Lines 284-285: Citations should be given for some negative studies.

Other points
13. The format of presenting numbers, decimals, ORs, CIs, etc. is very heterogeneous in the manuscript (commas vs. full stops, parentheses vs. brackets, etc.). This creates confusion. These aspects should be harmonized throughout the manuscript, including the Figures/Tables.
14. Lines 64-65: Only the high cost and the limited availability are real limitations. Performance is excellent and the inconvenience of an FNA is minimal.
15. The 15 studies that were analyzed should be listed and cited specifically (for example, in a Table). Did all of these studies analyze only FTCs/FTAs?

Round 2

Reviewer 3 Report

Major points:

1. Simple summary, Abstract and main text (lines 415-416): “Sonographic features associated with follicular lesions' malignancy are distinct from those widely reported for all thyroid cancers, including papillary thyroid cancer.” I cannot accept this conclusion, because it is not supported by the data. In fact, the immediately preceding sentence in the Abstract states that: “the most crucial features associated with an increased risk of FTC are: tumor protrusion (odds ratios – OR = 10.19), microcalcifications or mixed type of calcifications: 6.09, irregular margins: 5.11, marked hypoechogenicity: 4.59, and irregular shape: 3.6.” Except for tumor protrusion, all the other features are well-known to be associated with increased risk for thyroid carcinoma, including for PTC. The conclusion is thus inaccurate and misleading, and it should be deleted. Tumor protrusion can be discussed independently, without this inaccurate aphorism.

2. The reference to elastography and tridimensional Doppler is in my opinion irrelevant and should be deleted “Another promising method potentially differentiating FTA and FTC are elastography and tridimensional Doppler (21, 22).” I am not sure what “tridimensional Doppler” refers to here; reference 21 does not use Doppler, and reference 22 uses Doppler but makes no reference to “tridimensional”.

Minor point:

3. Title : I would delete « results of ».

Round 3

Reviewer 3 Report

All my comments have been properly addressed. I have no further comments.